# Inflammatory Bowel Disease and Reproductive Health: From Fertility to Pregnancy—A Narrative Review

**DOI:** 10.3390/nu14081591

**Published:** 2022-04-12

**Authors:** Camilla Ronchetti, Federico Cirillo, Noemi Di Segni, Martina Cristodoro, Andrea Busnelli, Paolo Emanuele Levi-Setti

**Affiliations:** 1Department of Biomedical Sciences, Humanitas University, Via Rita Levi Montalcini 4, 20090 Milan, Italy; camilla.ronchetti@humanitas.it (C.R.); federico.cirillo@humanitas.it (F.C.); noemi.disegni@humanitas.it (N.D.S.); martina.cristodoro@humanitas.it (M.C.); andrea.busnelli@humanitas.it (A.B.); 2Division of Gynecology and Reproductive Medicine, Fertility Center, Department of Gynecology, IRCCS Humanitas Research Hospital, Via Manzoni 56, 20089 Milan, Italy

**Keywords:** inflammatory bowel diseases, Crohn’s disease, ulcerative colitis, malabsorption, gut microbiota, inflammation, diet, infertility, pregnancy, breastfeeding

## Abstract

Despite the fact that knowledge on obstetrical management of Inflammatory Bowel Diseases (IBDs) has greatly improved over the years, many patients still actively avoid pregnancy for fear of adverse maternal or neonatal outcomes, of adverse effects of pregnancy on the disease activity, of eventual IBD inheritance, or of an increased risk of congenital malformations. Indeed, though data prove that fertility is hardly affected by the disease, a reduced birth rate is nevertheless observed in patients with IBD. Misconceptions on the safety of drugs during gestation and breastfeeding may influence patient choice and negatively affect their serenity during pregnancy or lactation. Moreover, physicians often showed concerns about starting IBD medications before and during pregnancy and did not feel adequately trained on the safety of IBD therapies. IBD-expert gastroenterologists and gynecologists should discuss pregnancy and breastfeeding issues with patients when starting or changing medications in order to provide appropriate information; therefore, pre-conception counselling on an individualized basis should be mandatory for all patients of reproductive age to reassure them that maintaining disease remission and balancing the eventual obstetrical risks is possible.

## 1. Introduction

The definition of Inflammatory Bowel Diseases (IBDs) includes two main multifactorial diseases: Crohn’s Disease (CD) and Ulcerative Colitis (UC). Both imply a chronic intermittent intestinal inflammation that can worsen due to an ineffective epithelial barrier function, triggers from the environment, genetic susceptibility, or an impaired immune reaction to the gut microbiota. Based on a recent epidemiological study, the incidence of IBD is increasing worldwide, due to the global evolution of newly industrialized countries developing in a modern, western-like manner [1]. As prevalence exceeds 0.3%, the increasing social burden highlights the need for research into prevention and improvement in health-care systems to manage these complex and costly diseases [2]. Since IBDs have their peak incidence at reproductive age (i.e., from 15 to 40 years of age [3]), and in half of patients, their diagnosis occurs before the age of 35 [4], their relationship with fertility and reproduction is crucial. IBD patients seem to have fewer children than the general population [5] despite similar fertility rates: the reasons for this voluntary childlessness could be anxiety about fertility, potentially adverse pregnancy outcomes, and the fear of vertical transmission of the disease to the offspring. Sexual dysfunction related to psychological comorbidities, like anxiety and depression, and body image misperception may also play an additional role [6].

It is therefore indisputably worthwhile to provide a practice-based review on the impact of IBD on fertility, pregnancy, and possible therapeutic strategies, in order to set useful practical guidelines both for physicians and patients. Once pharmaceutical and surgical options are well established, we will then focus our attention on diet and microbiomes, which are still new matters of debate.

## 2. IBD and Fertility

According to the latest European Crohn’s and Colitis Organization (ECCO) guidelines on reproduction, ulcerative colitis without previous pelvic surgery and inactive CD do not impair fertility [7]. Conversely, active CD may impair fertility via multiple factors such as fallopian tube inflammation and ovarian reserve lowering [6]. Different considerations should be made in UC patients who underwent ileal pouch anal anastomosis (IPAA), which seems to increase the risk of infertility by approximately threefold [8,9], mainly due to tubal dysfunction caused by adhesions.

On the other hand men with IBD may suffer from infertility due to two iatrogenic pathways: either the use of sulphasalazine producing reversible oligospermia or possible complications of IPAA (e.g., retrograde ejaculation or erectile dysfunction [7]).

Considering all the above possible factors lead to infertility, patients with IBD may be referred to Assisted Reproductive Technologies (ART) earlier than the general population, even after only six months of attempts [6]. It is still not clear if the ART success rate in IBD patients differs from the general population.

### 2.1. Active CD and Infertility

Although the mechanism of tubal damage by a chronic pelvic inflammatory status can be easily understood, evidence concerning the pathophysiological pathway that may diminish the ovarian reserve is still a matter of debate. The assessment of the ovarian reserve includes both biochemical analysis (i.e., basal Follicle Stimulating Hormone (FSH), estradiol and Anti-Mullerian Hormone (AMH) concentrations) and ultrasound imaging of the ovary for antral follicular count [10]. AMH can be measured at any point of the menstrual cycle and it is not an operator-dependent variable, which is the reason why it is commonly recognized as the best tool for an assessment of the ovarian reserve [11].

It was hypothesized that pelvic inflammation in CD patients could be related to a reduction of AMH levels; however, this correlation is still not clear. Senates et al. conducted a cross-sectional case–control study including 35 CD patients which revealed that AMH levels of affected women were significantly lower compared to healthy controls (1.02 ± 0.72 ng/mL vs. 1.89 ± 1.80 ng/mL, *p* = 0.009). Furthermore, among CD patients, those with active disease had lower levels compared to patients in remission. It is questionable whether this finding is applicable to the general affected population, or to a selected sub population, and different cohort studies tried to identify an age-related threshold. Notably, Frèour et al. found a significant reduction of AMH, but only in CD patients over 30 years of age, and showed an association between colonic localization of the disease and a lower ovarian reserve. On the other hand, according to a more recent study, the threshold age could be identified at 25 years [12]. Despite the evidence reported so far, a case control study conducted on 50 CD patients and published in 2021 shows that a decrease in AMH levels is detectable only in patients older than 30 years and with a disease duration equal or superior to 5 years [13].

In conclusion, it appears that an impact on ovarian reserve is evident, but only in some specific circumstances, such as long lasting and active disease; thus in order to reassure young women affected by CD, further studies or a meta-analysis is needed.

### 2.2. IPAA: Surgical Approaches and Effects

Almost 20 to 30% of patients with UC will necessitate colectomy despite medical therapy [14]. IPAA is the most used surgical treatment for these patients and it may also be required in the case of indeterminate colitis (IC) or CD [15,16].

IPAA can be successfully performed using a traditional laparotomic approach or by minimal invasive laparoscopy: ECCO guidelines on reproduction suggest the use of a laparoscopic approach. Indeed Beyer-Berjot et al. found a lower infertility rate in 63 patients who underwent laparoscopic IPAA compared with the infertility rate reported in previous studies, which were conducted on patients who underwent laparotomy (27% vs. 63%). Another study carried out on 160 women, through a questionnaire addressing medical and fertility history, observed a higher pregnancy rate after laparoscopic IPAA; however, a recent Cochrane review criticizes the latter study because of the high overall risk of bias and imprecise estimates [17].

### 2.3. IBD and ART Efficacy

From a nationwide Danish cohort study, it emerged that patients with UC and CD have a lower chance of live birth per cycle compared to infertile women without IBD. Notably, CD patients who had previous surgery had the worst prognosis, whereas IPAA appears not to have a negative impact [18]. According to the same group, prescribing corticosteroids in IBD patients before embryo transfer would improve their chances of a live born child [19]. Other studies found a comparable live birth rate between IBD patients and infertility controls [20,21]. It should be noted that these studies included a limited number of patients compared to the nationwide Danish cohort, and for this reason, in our opinion, in the absence of other large-scale studies, a worse ART prognosis should be suspected for IBD patients. Another debated matter is whether ART pregnancy outcomes differ in IBD patients. Norgard et al. reported a higher risk of preterm birth in UC patients, but the risk vanished when only singleton pregnancies were included [18]. Conversely, a recent study found a comparable pregnancy outcome between IBD patients, other infertile women undergoing ART, and patients with IBD with spontaneous conception [22].

## 3. IBD and Pregnancy

During pregnancy, many physiological changes occur in order to allow implantation and fetal growth. This is the reason why pregnancy represents a period of intense endocrine fluctuation and immune modulation [23]. In previous years, it was thought that during pregnancy there was a rise in maternal immune tolerance; however, it is now emerging that immunological states fluctuate during these months to meet various needs. For example, during the first stage of pregnancy (i.e., implantation), a proinflammatory Th1 state is predominant. Later, a tolerogenic Th2 response prevails until shortly before delivery when a new Th1 polarization occurs [24].

It is unclear if these placental immunological shifts are also associated with peripheral changes in the immune system response, which may, for example, affect disease activity in the intestinal tract. Studies published so far reported conflicting results [25,26,27]. It is well known that some autoimmune diseases improve during pregnancy (known as “the honeymoon period”), suggesting a potential different immune response that is also in peripheral sites [24,25,28]. More specifically, there would be an improvement of Th1-driven diseases [29]. CD is known to be a Th1/Th17 disease, whereas UC is more Th2/Th17 driven, and this would explain why the two diseases behave differently during pregnancy [30].

A recent study comparing IBD and non-IBD pregnant women [31] showed an improvement in the modulation of cytokine patterns during pregnancy in the first group. Indeed, IL-6, IL-8, IL-12, IL-17, and TNF-a proinflammatory cytokines significantly decreased after conception. During pregnancy itself, serum cytokine levels in patients with IBD subsequently remained relatively stable over the 40 weeks of gestation. On the other hand, Kim et al. [32] showed that a surrogate marker of bowel inflammation (i.e., Fecal Calprotectin (FC)), had higher levels in pregnant patients affected by IBD compared with controls, but it gradually decreased in the case-group. The opposite trend was observed in the control-group, demonstrating a slight gradual increase in their FC inflammation marker levels during gestation. As for babies born to mothers with IBD, the same study showed significantly higher FC levels compared with control babies from 2 to 36 months of age, after adjusting for sex, breastfeeding or not, antibiotic use, and delivery mode. The authors speculated that those babies may have been less able to achieve a balanced mucosal immunity or to establish an optimal intestinal barrier function. This fact is probably explained by a lower immune tolerance to commensal bacteria in babies born to IBD mothers, potentially leading to chronic mild intestinal inflammation due to a modification in the intestinal microbiota.

### 3.1. Medical Therapy

Safety data are available for most medications used in IBD so that their use is highly recommended to keep the disease in remission status.

Aminosalicylates (mesalazine, sulfasalazine, balsalazide, olsalazide), showed no increased obstetrical risk [33]. Formulations without dibutylphthalate are preferable in order to avoid any eventual risk of teratogenicity, even if this effect has not been proved in humans. The recommendations are to maintain pre-pregnancy doses and, if using sulfasalazine, to supplement with folate (>2 mg/day).

Corticosteroids, in view of their rapid conversion into less active metabolites, reach fetal blood in low and harmless concentrations [34]. Due to some concerns about teratogenic effects, such as cleft lip or palate, though it has not been confirmed [35], corticosteroids are recommended only in case of active flares.

In accordance with ECCO guidelines [7], the antibiotics metronidazole and ciprofloxacin should be prescribed only after the first trimester, due to the same risk of eventual orofacial malformations as corticosteroids [36].

As for immunomodulators, thiopurines (azathioprine or 6- mercaptopurine) are low-risk therapies in pregnancy [37], even if they showed a slight increase in preterm deliveries [38]. They are usually recommended as monotherapy in pregnant patients. In contrast to the above, methotrexate is a high-risk medication due to its strong teratogenic and abortive effects, so it must be discontinued at least 3 months before conception [39]. Lastly, in the immunomodulators group, cyclosporine should be used only as a rescue therapy for acute severe steroid-refractory UC, since no data on pregnant IBD women treated with cyclosporine are available [40].

Some anti-TNFα agents, such as infliximab, adalimumab and golimumab, cross the placenta, while certolizumab pegol does not. Even if they are considered safe in pregnancy, ECCO guidelines suggest stopping anti-TNF therapy around the 24th week of gestation to minimize transplacental transfer [7] if the patient’s case permits, because it is known that an early discontinuation may lead to a relapse of the IBD.

Nowadays, therapies with monoclonal antibodies have been routinely introduced to control IBD evolution. Unfortunately, during pregnancy, they should be avoided due to their transmission across the placenta and only a few may still be considered on a personalized decision basis. Among the latter, vedolizumab, the human IgG1 antibody against α4β7 integrin, and ustekinumab, the human IgG1 antibody that inhibits the p40 subunit of IL-12 and IL-23, are the only ones that can eventually be prescribed during gestation, but only as a last resort due to the lack of data from randomized controlled trials [40]. On the other hand, tofacitinib, the janus kinases 1 and 3 inhibitors, filgotinib and upadacitinib, both janus kinase 1 inhibitors, and ozanimod, the sphingosine 1-phosphate receptor modulator, are contraindicated due to the complete lack of data on pregnant women.

Drug safety and recommendations in pregnancy are listed in Table 1.

### 3.2. The Role of Diet during Pregnancy in IBD Patients

Diet and nutrition have an important role for patients with IBD. Different studies investigated the association between IBD and diet. On one hand, diet influences the evolution of the disease [41,42,43,44], on the other, IBD determines food preferences in patients in order to reduce disease activity [45].

Diet had been studied thoroughly in terms of macro- and micronutrient content, as well as food quantity and calorie intake.

Maternal dietary habits not only play a role in IBD activity, but also in fetal development and pregnancy.

The reduction of food intake to minimize symptoms is the primary cause of common malnutrition in IBD patients, with or without pregnancy. On the other hand, malnutrition is also a result of complications of the disease, such as diarrhea and bowel inflammation [45,46,47]. IBD mothers are particularly prone to malnutrition during the placental development stage [48,49], and this fact may determine nutritional deficiencies, leading to an increase of the probability of inadequate Gestational Weight Gain (GWG). It was demonstrated that IBD mothers with low GWG have a higher risk of preterm birth or of Small for Gestational Age (SGA) babies compared with healthy mothers with inadequate GWG or compared with IBD mothers without low GWG [50].

The avoidance of dairy sources is one of the factors that determines a reduced protein intake in IBD mothers, particularly in patients with Crohn’s disease. Intuitively, a low Proportion of Protein from Dairy Sources (PPDS), and the resulting low GWG, leads to an increased risk of SGA. Counterintuitively, Bengtson, Haugen et al. demonstrated the opposite theory: a low PPDS in IBD mothers is associated with a reduced risk of SGA babies, due to a possible unrecognized lactose malabsorption that frequently coexists with IBD [51]. In contrast with this thesis, other studies demonstrate that inadequate GWG is a strong predictor of SGA [52,53]. Low GWG can be either an independent risk factor for SGA or a marker of IBD activity. Importantly, pregnancy outcomes are not different in IBD patients with high GWG compared with IBD patients with adequate GWG [50].

Myklebust-Hansen, Aamodt et al. found that a traditional dietary pattern (characterized by fish, potatoes, rice, and cooked vegetable intake) is linked with a lower risk of SGA in IBD mothers [54]. Indeed, the literature is mainly focused on the protective effect of some food products on SGA and preterm delivery. In particular, the Mediterranean diet, based on fruits and vegetables, unrefined grains, olive oil, and fish is highly recommended [41,42,43,44]. Furthermore, a study within the Danish National Birth Cohort demonstrated that the Western diet determines a higher risk of SGA [41]. This association can be explained by the impact of diets on microflora and on bowel inflammation status.

In the Mediterranean diet, there are several nutrients important for fetal development, not only for IBD activity: nutritional deficiencies are associated with an altered transfer of nutrients through the placenta, and consequently, with adverse pregnancy outcomes [54]. Some of these nutrients are calcium, potassium, magnesium, zinc, folic acid, and protein [55,56]. An adequate consumption of calcium is particularly difficult in a dairy product-free diet [57].

As anticipated above, IBD patients’ diets are characterized by the absence of dairy products in order to avoid the discomfort of lactose malabsorption [45]. The unabsorbed sugar determines microflora fermentation, and consequently, production of gases and bacterial metabolites. This may be one of the pathological mechanisms of diarrhea in patients who suffer from IBD, but it is not the only one [51]. In fact, it was demonstrated that lactose intolerance, diagnosed by a breath hydrogen test, is not more frequent in IBD patients than in healthy controls [58,59]. The malabsorption is probably secondary to the bowel disease and its complications, such as inflammation, bowel resections, intestinal bowel overgrowth, and short gut transit time [60,61]. Moreover, diarrhea can also usually be correlated to Irritable Bowel Syndrome (IBS), which coexists in most cases with IBD [51,62].

Lastly, the requirement of nutrients strictly depends on the stage of the disease [56]. Particularly during the active phase of the disease, IBD patients prefer a traditional dietary pattern, in which raw fruits and vegetables and dairy products are avoided. In fact, the consumption of raw fruits and vegetables is frequently associated with the worsening of symptoms [63]. The transient lactose intolerance is the reason why dairy products are excluded during the active phase of the disease [45,64]. In conclusion, dietary patterns influence the disease activity and so, directly and indirectly, they also influence pregnancy outcomes [54].

### 3.3. The Role of Microbiome

In 2001, for the first time, Lederberg formulated the term “microbiome, to signify the ecological community of commensal, symbiotic, and pathogenic microorganisms that literally share our body space” [65]. Bringing this concept to a further level, each human being can be considered as an assembly of somatic cells, and of different and multiform symbiotic species. These microbial members play multiple roles, collaborating with the immune system, and contributing to the digestion of dietary substrates. Recently, updated techniques are emerging for the study of such species (i.e., metaproteonomics [66], metagenomics [67] and trancriptomics [68]). The modification of the host-associated microbial communities (the “microbiota”) has been related to some disorders such as obesity, malnutrition, or IBD [69], or to certain periods of life, for example pregnancy, since microbiomes and the immune system have a close reciprocal relationship [70,71,72]. Indeed, even in healthy gestations, microbial diversity decreases until the third trimester, reaching a pattern similar to the microbiome in patients with metabolic disease [73].

In addition, IBD patients appear to have reduced fecal bacterial diversity, with low concentrations of commensal bacteria producing butyrate (e.g., Faecalibacterium prausnitzii), and an abundance of Proteobacteria and Actinobacteria [74,75,76]. These findings are similar to what happens in a healthy pregnancy, particularly in the third trimester, during which the increase of Proteobacteria and Actinobacteria seems to contribute to the inflammatory environment necessary for delivery [77].

At birth, the human microbiota is homogeneously distributed across the body. The first determinant of a newborn’s bacterial community composition is the delivery mode. Children born by vaginal delivery are initially exposed to maternal vaginal, urogenital, or fecal microbiota as they pass through the birth canal [78,79]. On the other hand, children delivered by cesarean section are exposed to maternal skin microbiota [78].

Extraordinarily, the effects of the delivery mode may persist for long time and have consequences for the child’s future health status. For example, children born via cesarean section showed a higher risk for some immune-mediated diseases [80,81].

In early life, the different postnatal developmental factors that assemble in the human microbiota, play an important role in childhood health, providing resistance to pathogen invasion and immune stimulation [82]. For example IBDs, such as necrotizing enterocolitis, malnutrition [83], asthma [84], or allergies [85], have been linked to an altered postnatal microbiome acquisition, altered development, or to a low bacterial diversity in early life. Exclusive breast-feeding rather than formula feeding, fever status, or antibiotic therapy can induce environmental selection on infant microbiota too.

Weaning is the starting point for the onset of a transition towards the adult intestinal microbiome. Life events can still induce modifications, although specific compositions appear to recover and changes can be transient [86].

As for the microbiome in pregnant women with IBD, recent data suggest that, as immunological parameters improve in pregnancy, microbial diversity normalizes to that seen in healthy pregnant women [31]. On the other hand, children born to IBD mothers showed an altered gut microbiome correlated to abnormal adaptive immune systems [87].

In addition, Torres et al. detected a significant association between maternal IBD with changes in microbiome composition during gestation and in the offspring. Their data may provide a potential link between early life exposures, microbiome, and future risk of IBD.

As Cox et al. demonstrated for adult obesity, microbiota disruption during the critical early developmental window can explain later life phenomena, including metabolic consequences and inflammation [88]. Recently, various studies found an association between diet and microbiome. The MELODY (Modulating Early Life Microbiome through Dietary Intervention in Pregnancy) trial, for example, evaluated if diet during the third trimester of pregnancy could influence the microbiome of patients with IBD and of their babies to improve the immune system [89]. Understanding the critical events shaping the microbiome of IBD mothers and their children can surely provide new strategies for disease prediction and prevention [87].

## 4. IBD and Delivery

Although there are no guidelines on the mode of delivery in IBD patients, vaginal delivery is commonly suggested, especially if the disease is quiescent or mild.

On the other hand, in case of IPAA, cesarean section should be advised due to the higher risk of the alteration of sphincter pressure that could be more frequently damaged by vaginal deliveries.

Similarly, in the case of active perianal disease, a higher risk of severe sphincter injuries has been reported [90] and cesarean delivery should be preferred.

In conclusion, the decision on delivery mode should be taken considering both the severity of the perianal disease and previous surgeries in that area.

In case of ileostomy or colostomy, a multidisciplinary discussion must be held between the gynecologist, the colorectal surgeon, and the gastroenterologist, due to the lack of data on outcomes after deliveries. To be considered in such consultations, is the fact that in pregnancy, because of the growing uterus or the presence of post-surgical adhesions, an obstruction of the small bowel may occur in case of ileostomy, with consequent obstetrical complications [91].

## 5. IBD and Lactation

Hormonal changes are typical when starting breastfeeding, the lower compliance in taking therapies or the eventual resumption of the smoking habit may lead to a flare-up of IBD during postpartum. This fact, and the fear of possible adverse effects on the baby, may cause the avoidance of breastfeeding or a discontinuation of medical therapies.

Physicians have a key-role in a proper counselling to support breastfeeding with a special emphasis on possible benefits for both mother and newborn, and to educate women not to discontinue therapies, to avoid smoking, and to keep the disease monitored.

Indeed, during breastfeeding, many therapies are considered safe for the newborn and maternal milk is shown to protect babies from the development of an early-onset IBD [92].

Aminosalicylates (mesalazine, sulfasalazine, balsalazide and olsalazide), due to their minimal excretion in maternal milk [93], are considered safe and must be discontinued only in case of neonatal severe bloody diarrhea, though this is a very rare adverse event [94]. The use of thiopurines is equally advisable, since they did not demonstrate a higher risk of physical or developmental anomalies in newborns breastfed by mothers under such regimen therapy [95]. In order to have a lower amount of the drug in mothers’ milk, it is recommended to breastfeed babies 4 h after taking corticosteroids [96], 12–24 h after metronidazole, and 48 h after ciprofloxacin [97]. These different timings are explained by the different clearances of the drugs, and for this reason, a short-term antibiotic regimen should be preferred to a long-term one [98]. On the other hand, a proved contraindication is established for the use of methotrexate and cyclosporine since they are both excreted in large amounts in breast milk, and may cause immune suppression, neutropenia, and carcinogenesis in breastfed infants, or potential alterations in cellular metabolism, respectively [99]. Infliximab, adalimumab, and certolizumab can be prescribed during lactation because they are only transferred in breast milk in small amounts [100], and their molecules are disactivated by neonatal digestion enzymes. As for other, more recent therapies, safety data are still missing, so their use is not recommended.

Drug safety and recommendations in breastfeeding are listed in Table 1.

## 6. Conclusions

Despite the fact that knowledge on the obstetrical management of IBD has greatly improved over the years, many patients still actively avoid pregnancy for fear of adverse maternal or neonatal outcomes, of the adverse effect of pregnancy on disease activity, of eventual IBD inheritance, or of an increased risk of congenital malformations [101]. Indeed, though data prove that fertility is hardly affected by the disease, [3] a reduced birth rate is nevertheless observed in patients with IBD [102]. Misconceptions on the safety of drugs during gestation and breastfeeding may influence patient choice and negatively affect their serenity during pregnancy or lactation. Moreover, physicians often showed concerns about starting IBD medications before and during pregnancy and did not feel adequately trained on the safety of IBD therapies. IBD-expert gastroenterologists and gynecologists should discuss pregnancy and breastfeeding issues with patients when starting or changing medications in order to provide appropriate information; therefore, pre-conception counselling on an individualized basis should be mandatory for all patients of reproductive age to reassure them that maintaining disease remission and balancing the eventual obstetrical risks is possible.

Practical take home messages for IBD patients of reproductive age are summarized in Table 2, and a flowchart to guide both physicians and patients in young women affected by IBD management is shown in Figure 1.

## Figures and Tables

**Figure 1 nutrients-14-01591-f001:**
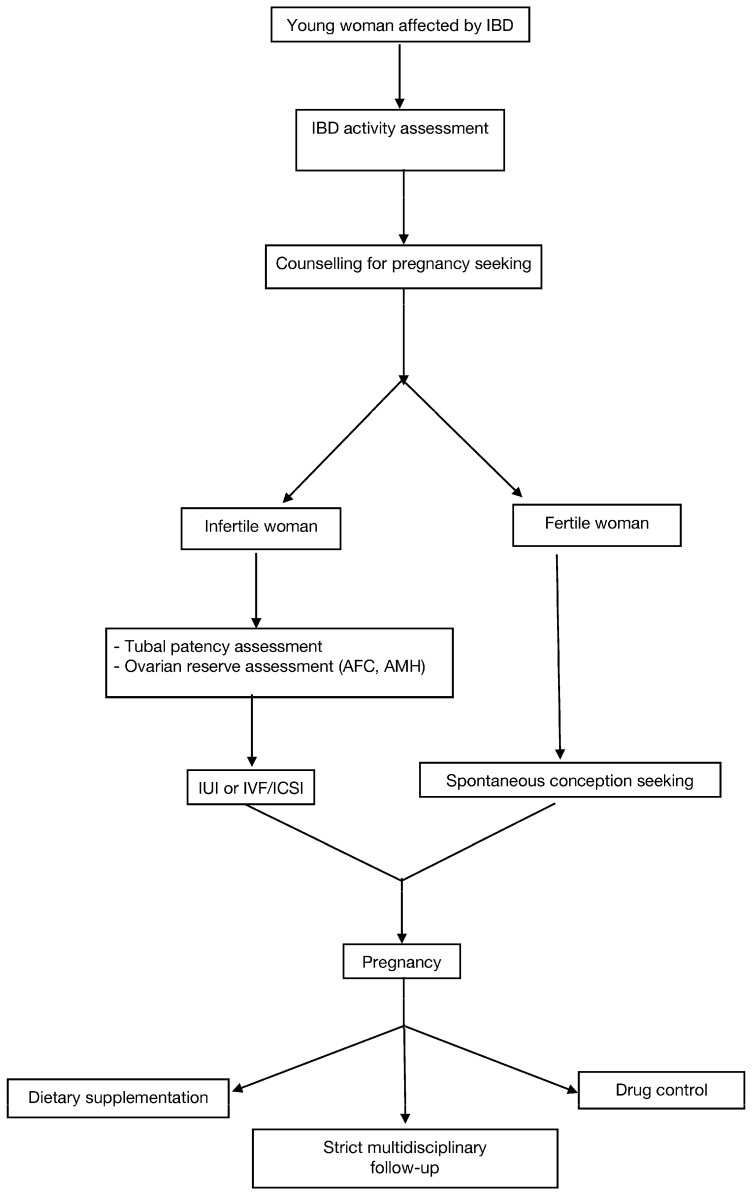
Management flowchart for young women affected by Inflammatory Bowel Disease at reproductive age (IBD: Inflammatory Bowel Disease; AFC: Antral follicular count; AMH: Anti-Mullerian Hormone; IUI: Intrauterine Insemination; IVF: In Vitro Fertilization; ICSI: Intracytoplasmic Sperm Injection).

**Table 1 nutrients-14-01591-t001:** Drug safety and recommendations in pregnancy and breastfeeding.

Medical Treatment	Safety and Recommendations in Pregnancy	Safety and Recommendations in Breastfeeding
Aminosalicylates (mesalazine, sulfasalazine, balsalazide, olsalazide)	No increased obstetrical risk. Always recommended (formulation without dibutylphthalate are preferable and, if sulfasalazine is used, suggestion to supplement with folate).	Safe and must be discontinued only in case of neonatal severe bloody diarrhea.
Corticosteroids	Concerns about teratogenic effects, such as cleft lip or palate. Recommended only in case of active flares.	Recommended to breastfeed babies 4 h after taking corticosteroids.
Antibiotics (metronidazole and ciprofloxacin)	Concerns about teratogenic effects, such as cleft lip or palate. Recommended only after the first trimester of gestation.	Recommended to breastfeed babies 12–24 h after metronidazole and 48 h after ciprofloxacin intake. A short-term antibiotic regimen must be preferred.
Thiopurines (azathioprine or 6- mercaptopurine)	Slight increase in preterm deliveries. Recommended as monotherapy.	Advisable, no a higher risk of physical or developmental anomalies in newborns.
Methotrexate	Strong teratogenic and abortive effects. Never recommended in pregnancy.	Contraindicated.
Cyclosporine	No data on pregnant women available, only recommended as rescue therapy for acute severe steroid-refractory ulcerative colitis.	Contraindicated.
Anti-TNFα agents (infliximab, adalimumab, golimumab and certolizumab)	Evidence of crossing the placenta, except of certolizumab. Recommended stopping around the 24th week of gestation, if the case permits.	Safe due to their transmission in breast milk only in small amounts and deactivation by neonatal digestion enzymes.
Vedolizumab and ustekinumab	Should be avoided due to their transmission across the placenta and partial lack of data in pregnancy. Can eventually be prescribed only as an ultimate alternative.	Safety data are still missing, so their use is not recommended.
Tofacitinib, filgotinib and upadacitinib	Contraindicated due to the complete lack of data in pregnancy.	Safety data are still missing, so their use is not recommended.

**Table 2 nutrients-14-01591-t002:** Practical take home messages for Inflammatory Bowel Diseases (IBD) patients in reproductive age.

Clinical Scenario	Practical Take Home Messages
IBD and female fertility	Ulcerative colitis without previous pelvic surgery and inactive Crohn’s disease (CD) do not impair fertility. On the other hand, active CD may impair fertility via multiple factors such as fallopian tube inflammation and a lowering of the ovarian reserve. Ileal pouch anal anastomosis (IPAA) seems to increase the risk of infertility by approximately threefold, mainly because of a tubal dysfunction caused by adhesions.
IBD and male fertility	Men with IBD may suffer from infertility due to two iatrogenic pathways: either the use of sulphasalazine producing reversible oligospermia or possible complications of IPAA (e.g., retrograde ejaculation or erectile dysfunction).
IBD and Assisted Reproductive Technologies (ART) efficacy	IBD patients may be addressed to ART earlier than the general population even after only six months of attempts. It is still not clear if the ART success rate in IBD patients differs from the general population.
IBD and pregnancy	IBD pregnant patients show an improvement in the modulation of cytokine patterns during pregnancy and a gradual decrease of inflammation marker levels during gestation. As for babies born to mothers with IBD, they may have a lower ability to achieve a balanced mucosal immunity or establish an optimal intestinal barrier function, a fact that may lead to a higher risk of IBD recurrence in the offspring.
IBD and diet in pregnancy	Diet influences the evolution of the disease and IBD determines food preferences in patients in order to reduce disease activity. Fruits and vegetables have a protective effect; on the contrary, carbohydrates, fats, and dairy products should be avoided in these patients. Malnutrition is the result of the complications of the disease, but also of self-reduction of food intake in order to minimize the symptoms. Malnutrition determines nutritional deficiencies, leading to an increase of the probability of inadequate gestational weight gain, with a consequent higher risk of preterm birth or small for gestational age babies.
IBD and microbiome in pregnancy	IBD patients seem to have reduced fecal bacterial diversity with a low concentration of commensal bacteria producing butyrate, and an abundance of Proteobacteria and Actinobacteria, which also happens in healthy pregnancies. In IBD patients, as immunological parameters improve in pregnancy, microbial diversity normalizes to that seen in healthy pregnant women. On the other hand, children born to IBD mothers showed an altered gut microbiome that is correlated to abnormal adaptive immune systems and a future risk of IBD.
IBD and delivery	Although there are no guidelines on the mode of delivery in IBD patients, vaginal delivery is commonly suggested, especially if the disease is quiescent or mild. On the contrary, in case of IPAA, a cesarean section should be advised due to the higher risk of the alteration of sphincter pressure that could be more frequently damaged by vaginal deliveries. Similarly, in case of active perianal disease, a higher risk of severe sphincter injuries has been reported, and cesarean delivery should be preferred.
IBD and lactation	Physicians have a key-role in a proper counselling to support breastfeeding, with a special emphasis on possible benefits for both mothers and newborns, and to educate women not to discontinue the therapy, to avoid smoking, and to keep the disease monitored. Indeed, during breastfeeding, many therapies are considered safe for the newborn, and maternal milk is shown to protect babies from the develop of an early-onset IBD.

## Data Availability

Not applicable.

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
