# Peer review of "Inflammatory Bowel Disease and Reproductive Health: From Fertility to Pregnancy—A Narrative Review"

_nutrients, 2022, doi:10.3390/nu14081591_

Round 1
Reviewer 1 Report
This is a very well researched review and covers IBD in perspective with fertility and pregnancy.Crohn's disease and ulceratice colitis are the majorily predominant diseases of IBD.The review is well segregated into pertinent topics and subtopics which do justice to the review.In particular IBD and fertility and IBD and pregnancy are covered in detail and provide sufficient information about patients and the related concerns.
The authors have compiled very useful tables namely drug safety and recommendations in pregnancy and brestfeeding and as well as practical take-home messages.These would be meaningful to clinicians as well as patients as well as researchers.
There are a few english language corrections which can be done.
Author Response
Thank you very much for your kind attention and positive comments. A
native English-speaking colleague checked our manuscript.
Kind regards,
the authors.
Reviewer 2 Report
This is a well-written review paper on the topic of inflammatory bowel disease and reproductive health.
Please pay attention to the way text is organized.
Part 2 - format references properly and in single style
L57 and on - check for excessive (double) spacing, unneccessary comas throughout a text. Use MDPI corrective service if necessary
Part 3.3 - even though influence of microbiome is well-covered, it will be beneficial to mention methods of study: Metaproteomics (https://www.mdpi.com/2076-2607/9/5/980) ; Metagenomics (https://www.mdpi.com/2076-2607/10/4/711) ; Transcriptomics (https://www.mdpi.com/2076-3417/12/5/2483).
Also, audience will appreciate several figures/schemes to ease absorbtion of material.
Overall, I believe manuscript worth publishing.
Best reagrds.
Author Response
We are glad that you appreciated our effort.
Thank you very much for your kind and wise comments and suggestions. We revised the paragraphs and subparagraphs’ numbering, formatted the references in part 2 and fully revised the spacing and punctuation. We added the kindly suggested topics and relative references in part 3.
At last, according to your kind advise, we added a flowchart-structured scheme to guide the management for young IBD affected patients in reproductive age.
Hoping we have improved our narrative review,
kind regards,
The authors.